# FGFR Fusions in Cancer: From Diagnostic Approaches to Therapeutic Intervention

**DOI:** 10.3390/ijms21186856

**Published:** 2020-09-18

**Authors:** Antonella De Luca, Riziero Esposito Abate, Anna Maria Rachiglio, Monica Rosaria Maiello, Claudia Esposito, Clorinda Schettino, Francesco Izzo, Guglielmo Nasti, Nicola Normanno

**Affiliations:** 1Cell Biology and Biotherapy Unit, Istituto Nazionale Tumori-IRCCS-Fondazione G. Pascale, 80131 Naples, Italy; a.deluca@istitutotumori.na.it (A.D.L.); r.espositoabate@istitutotumori.na.it (R.E.A.); am.rachiglio@istitutotumori.na.it (A.M.R.); m.maiello@istitutotumori.na.it (M.R.M.); claudia.esposito@istitutotumori.na.it (C.E.); 2Clinical Trials Unit, Istituto Nazionale Tumori-IRCCS-Fondazione G. Pascale, 80131 Naples, Italy; c.schettino@istitutotumori.na.it; 3Division of Surgical Oncology, Hepatobiliary Unit, Istituto Nazionale Tumori-IRCCS-Fondazione G. Pascale, 80131 Naples, Italy; f.izzo@istitutotumori.na.it; 4SSD Innovative Therapies for Abdominal Cancers, Istituto Nazionale Tumori-IRCCS-Fondazione G. Pascale, 80131 Naples, Italy; g.nasti@istitutotumori.na.it

**Keywords:** fibroblast growth factor receptors, FGFR fusions, next generation sequencing, cancer, FGFR inhibitors

## Abstract

Fibroblast growth factor receptors (FGFRs) are tyrosine kinase receptors involved in many biological processes. Deregulated FGFR signaling plays an important role in tumor development and progression in different cancer types. *FGFR* genomic alterations, including *FGFR* gene fusions that originate by chromosomal rearrangements, represent a promising therapeutic target. Next-generation-sequencing (NGS) approaches have significantly improved the discovery of *FGFR* gene fusions and their detection in clinical samples. A variety of FGFR inhibitors have been developed, and several studies are trying to evaluate the efficacy of these agents in molecularly selected patients carrying *FGFR* genomic alterations. In this review, we describe the most frequent *FGFR* aberrations in human cancer. We also discuss the different approaches employed for the detection of *FGFR* fusions and the potential role of these genomic alterations as prognostic/predictive biomarkers.

## 1. Introduction

Fibroblast growth factor receptors (FGFRs) are highly conserved tyrosine kinase receptors that play an important role in human cancer. Following the binding of growth factors of the fibroblast growth factor (FGF) family, FGFRs dimerize and activate intracellular signaling pathways responsible of cellular proliferation and survival [1]. The FGFR system regulates several crucial developmental processes, including the induction of organogenesis and morphogenesis, and homeostatic processes in adult tissues, such as repair and remodeling [2]. Aberrant activation of FGFRs is observed in different cancer types and plays a role in tumor development and progression. Deregulated FGFR signaling results from different mechanisms. Overexpression of FGFRs might occur either in the presence or absence of genetic alterations (i.e., gene amplification). Mutations (single nucleotide variants, SNVs) of *FGFRs* have been described in different tumor types. Aberrant activation of FGFRs might also be driven by autocrine and paracrine circuits supported by increased synthesis and release of FGFR ligands [3]. Chromosomal rearrangements leading to *FGFR* gene fusions have been also found to be involved in the pathogenesis of human cancer.

Gene fusions are hybrid genes that originate from the chromosomal rearrangement of two genes, in the form of translocation, insertion, inversion, and deletion [4]. Fusion events, which involve a variety of partner genes, result in the formation of fusion proteins capable of oncogenic transformation and induction of oncogene addiction. The discovery of targetable fusions and the improvement of techniques used for detecting these alterations allowed the development of specific therapies for the treatment of fusion-driven tumors [5]. 

The growing therapeutic relevance of *FGFR* alterations, including fusions, in different cancer types has greatly supported the development of a variety of novel agents along with the improvement of diagnostic tests. In this review, we will focus on the biology of the FGFR system and on the frequency of *FGFR* aberrations in human cancer. We will also describe the different approaches employed for the detections of fusions and the potential role of these genomic alterations as prognostic/predictive biomarkers.

## 2. The FGFR/FGF System

The FGFR family comprises four highly conserved tyrosine kinase receptors (RTKs): FGFR1, FGFR2, FGFR3, and FGFR4, consisting of three extracellular immunoglobulin (Ig)-type domains (D1–D3), a single transmembrane domain, and a cytoplasmic tyrosine kinase domain [6]. A unique characteristic of FGFRs is the presence of an acidic, serine-rich sequence, termed the acid box, in the linker region between D1 and D2. The D2–D3 region is necessary for ligand binding and specificity. The D1 domain and the acid box seem to play a role in FGFR autoinhibition [7]. A fifth member of the FGFR family has been discovered, termed fibroblast growth factor receptor-like 1 (FGFRL1/FGFR5), which interacts with heparin and FGF ligands [8]. Like the other members of the FGFR family, FGFR5 consists of three extracellular Ig-like domains and a single transmembrane helix, but it lacks the tyrosine kinase domain, which is replaced by a short intracellular tail with a peculiar histidine-rich motif [9]. The biological function of FGFR5 is unclear. A recent study suggested that it functions as a cell–cell adhesion protein, acting as a tumor suppressor gene [10]. 

Alternative splicing in the D3 domain of *FGFR1*, *2,* and *3*, generates isoforms IIIb and IIIc with different FGF-binding specificity. The IIIb isoforms are predominantly expressed in epithelial tissues, whereas IIIc isoforms are expressed in mesenchymal tissues. Alternative splicing and switching from epithelial to mesenchymal isoforms are involved in the epithelial-to-mesenchymal transition and in tumor progression [11]. In this regard, genomic rearrangements leading to the generation of fusion proteins might also alter the splicing of *FGFR* isoforms. However, no data on the involvement of this phenomenon in the growth of cancer addicted to *FGFR* fusions are available. Soluble splice variants of FGFR4 have been recently described, although further studies are required to better define the biological functions of these isoforms [12,13].

The FGF family of proteins is composed of 18 ligands (FGF1–FGF10 and FGF16–FGF23). Members of five of the six subfamilies act as paracrine factors, whereas members of the FGF19 subfamily (FGF19, FGF21, and FGF23) work in an endocrine fashion [7]. Four FGF homologous factors (previously indicated as FGF11–FGF14) fail to activate any FGFRs and are not considered members of the FGF family [14], whereas FGF15 is the mouse orthologue of FGF19. FGF ligands interact with heparan sulfate proteoglycans that are present both at the cell surface and in the pericellular and extracellular matrix. Heparan sulfate proteoglycans are obligatory cofactors of paracrine FGFs for FGFR activation, whereas endocrine FGFs preferentially require Klotho proteins as co-receptors to initiate FGFR signaling [15].

Ligand binding to the receptor induces FGFR dimerization and the subsequent phosphorylation of the tyrosine kinase domain. Activation of the receptor promotes the phosphorylation of intracellular substrates, including FGFR substrate 2α (FRS2α) and phospholipase Cγ1 (PLCγ1). FRS2α activates RAS/MEK/ERK and PI3K/AKT signaling pathways that regulate cell proliferation and survival, whereas PLCγ1 stimulates cell motility through the activation of protein kinase C (PKC) and calcium-dependent proteins [2]. Other pathways are activated by FGFRs, including JAK/STAT, p38MAPK, Jun N-terminal kinase, and RSK2 [16]. Different negative regulators, including Sprouty proteins and MAPK phosphatase 3 attenuate FGFR signaling [6].

## 3. Genetic Alterations of FGFRs in Human Cancers

Deregulated FGFR signaling is observed in various tumor types. A recent study that analyzed the *FGFR* genomic alterations in 4853 tumor samples by next-generation sequencing (NGS), described the presence of *FGFR* alterations in 7.1% of cases [17]. Genetic aberrations of *FGFR1* are more frequently observed in human cancers (2.86%), followed by alterations in *FGFR3* (2.21%), *FGFR2* (1.77%), and *FGFR4* (1.54%).

### 3.1. *FGFR* Amplifications and Mutations

Gene amplifications are the most frequent *FGFR* alterations reported in human cancers accounting for 66% of all *FGFR* aberrations [17]. Gene amplification often leads to the overexpression of FGFR proteins, resulting in the aberrant activation of the receptors and an increased downstream signaling [1]. *FGFR1* is the most commonly amplified gene (2.25%) [18]. *FGFR1* amplification is frequently observed in breast, lung, and colon cancer [18]. *FGFR2* amplification is less frequent (0.34%) and has been described in some cancer types, including breast, gastric, and esophageal carcinoma [18]. *FGFR3* gene amplification (0.31%) has been observed in breast carcinoma, bladder carcinoma, glioblastoma multiforme, pancreatic cancer, and lung adenocarcinoma. *FGFR4* amplification is rare (0.16%).

*FGFR* mutations are less frequent than *FGFR* amplifications, representing 26% of the aberrations detected in *FGFR*-altered tumors [17]. *FGFR* mutations can affect the extracellular or the transmembrane or the kinase domains of FGFRs and result in a deregulated FGFR signaling through various mechanisms, including the increased kinase activation, the reduced degradation of the receptor, or the abnormal receptor dimerization [1,16]. Mutations in *FGFR1* have been observed in 1.12% of cases, with a prevalence in lung, colon, breast, endometrial adenocarcinoma, and glioblastoma multiforme. The most frequent *FGFR1* activating mutation is the N546K (0.12%) [18] in the kinase domain of the receptor that alters the tyrosine auto-phosphorylation with an increased kinase activation [19]. Mutations in *FGFR2* and *FGFR3* are more frequent (1.36% and 1.83%, respectively) [18]. The most common *FGFR2* activating mutations are the S252W mutation in the extracellular domain (0.17%), the N549K mutation in the tyrosine kinase domain (0.06%), and the C382R mutation affecting the transmembrane domain of the receptor (0.06%) [18]. The most frequent *FGFR3* activating mutation is the S249C missense mutation that resides in the extracellular domain of the receptor (0.54%) [18]. This mutation induces ligand-independent dimerization and constitutive phosphorylation of the receptor [20]. The *FGFR3* S249C mutation is relatively frequent in bladder cancer (66.6%) [21]. *FGFR4*-activating mutations are rare and are detected in some pediatric tumors, such as rhabdomyosarcoma [22]. A novel oncogenic mutation of *FGFR4* (G636C) has been recently discovered in gastric cancer [23].

### 3.2. *FGFR* Family Gene Fusions

*FGFR* fusions have been described in several tumor types, although the incidence is low (8%) [17]. *FGFR* fusions can be classified into type I or type II fusions. In type I fusions, where the *FGFR* is the 3′ fusion partner, the extracellular and the transmembrane domains are excluded from the fusion protein, which includes only the *FGFR* kinase domain linked to the 5′ protein partner. In type II fusions, with the *FGFR* as the 5′ fusion gene, the breakpoint usually occurs in exons 17, 18, or 19, and the extracellular, the transmembrane, and the kinase domain remain intact [16]. In both types of fusion proteins, the diverse FGFR fusion partners contribute with specific domains that favor the dimerization, including the coiled-coil, the SPFH, the sterile alpha motif (SAM), the LIS1-homologous (LisH), the IMD, and the caspase domains [24]. Such ligand-independent increased dimerization provides oncogenic potential to the FGFR fusion protein. Fusion genes between *FGFR1–2–3* and multiple partners have been identified in several tumor types (Table 1). It is rare to find *FGFR* fusions together with *FGFR* mutations, suggesting that the presence of unique alterations is sufficient to drive cancer progression.

*FGFR1* fusions are rare in solid tumors. A *FGFR1–HOOK3* gene fusion has been observed in gastrointestinal stromal tumor (GIST) [25]. *FGFR1–TACC1* was detected in GIST, in grade II *IDH* wild-type glioma, and in glioblastoma [25,26,27], whereas *FGFR1–ZNF703* was detected in breast cancer [28]. These fusions involve the N-terminus of the FGFR1 protein and the coiled coil of the fusion partners to induce activation of the receptor and downstream signaling. The *FGFR1–NTM* fusion, whose functional effect is unknown, was detected in bladder urothelial carcinoma [17,29]. *BAG4–FGFR1* was identified in non-small cell lung cancer (NSCLC) [30,31].

*FGFR2* fusions are the most frequent *FGFR* fusions [17]. As compared with the other member of the FGFR family, *FGFR2* had several reported partners and *FGFR2* fusions are particularly common in cholangiocarcinoma. In this regard, *FGFR2–AHCYL, FGFR2–BICC1, FGFR2–PPHLN1*, and *FGFR2–TACC3* fusions have been frequently described in patients with intrahepatic cholangiocarcinoma, although over 100 different *FGFR2* partners have been reported in this disease [32,33,34,35,36,37,38]. These fusions activate the canonical FGFR signaling and possess oncogenic activity [37,39]. The *FGFR2–CCDC6* fusion has been demonstrated to induce cancer cell proliferation and tumorigenesis in vivo [38,40]. Several other partners involved in *FGFR2* fusion genes, whose biological activity has not been fully characterized, have been described in cholangiocarcinoma, including *KIAA1217, KIAA1598, DDX21, LAMC1, NRAP, NOL4, PHC1, RABGAP1L, RASAL2, ROCK1, TFEC, AFF4, CELF2, DCTN2, DNAJC12, DZIP1, FOXP1, INA, KCTD1,LGSN, LPXN, MYPN, PRKN, PCM1, RNF41, SH3GLB1, STK3, SORBS1, TBC1D1,* and *UBQLN1* [33,34,35]. 

*FGFR2–BICC1* has been also identified in colorectal cancer and hepatocarcinoma, although with low frequency [32]. Two *FGFR2*–*KIAA1598* fusions and other *FGFR2* fusions with novel partners (*CIT, ERC1, LZTFL1, POC1B, SORBS1, TP73, TXLNA*) have been recently identified in a large cohort (n = 26054) of lung cancer patients [31].

*FGFR3* fusions are more commonly observed in glioblastoma, bladder, and lung cancer [18]. The majority of FGFR3 fusions are with transforming acidic coiled-coil 3 (TACC3) and result from the in-frame fusion of the FGFR3 N-terminus with the TACC3 C-terminus [27]. TACC3 protein has a coiled-coil domain at the C terminus and is involved in mitotic spindle assembly and stability [41]. *FGFR3–TACC3* fusions have been described in different tumor types, including glioma, lung cancer, bladder cancer, head and neck squamous cancer, lung squamous cell carcinoma, and cervical cancer [26,27,30,31,38,42,43]. The FGFR3–TACC3 fusion protein induces a constitutive activation of the tyrosine kinase domain with the consequent activation of MEK/ERK and STAT1 signaling, but not PLCγ1, as the tyrosine residue in exon 19, responsible for the interaction with PLCγ1, is lost [38,42]. The FGFR3–TACC3 protein also induces mitotic and chromosomal segregation defects and generates aneuploidy [27]. The presence of the FGFR3–TACC3 fusion increased the proliferation of cancer cell lines [27,38] and induced tumorigenesis in mice [27].

Among other *FGFR3* fusions, *FGFR3–BAIAP2L1* has been described in bladder and lung cancer [42,44]. *FGFR3–BAIAP2L1* fusion promotes the constitutive activation of FGFR3 signaling with a potent oncogenic activity [42,44]. Other fusion partners of *FGFR3* include *AES, ELAVL3, JAKMIP1, TNIP2,* and *WHSC1* [5,17].

Recently, *FGFR4* fusions (*ANO3–FGFR4, NSD1–FGFR4*) have been identified in NSCLC patients [31].

## 4. Approaches to Detect *FGFR* Fusions in Clinical Diagnostics

Since the discovery of the first chromosomal rearrangements in hematologic malignancies using chromosomal-banding techniques, technological advancements have enabled the detection of a wide number of gene fusions in many tumor types [4,5,45,46]. The development of fluorescence in situ hybridization (FISH) technique in combination with cytogenetics allowed the simultaneous visualization of different chromosome structures in different colors, significantly improving the localization of chromosomal breakpoints. This approach employs fluorescently labeled DNA probes that bind to specific complementary target sequences. Detection of the signals is performed by fluorescence microscopy [47]. In particular, the break-apart FISH assay allows the identification of gene translocations using probes specific for loci that are physically close in the wild-type configuration. The wild-type signal pattern shows two pairs of closely approximated or fused signals, whereas the two colors split apart when a translocation occur [48].

As compared to standard cytogenetics, FISH analyses do not require living cells; can be easily performed on clinical formalin-fixed, paraffin-embedded (FFPE) samples; and are a technique with a relatively fast turnaround time. However, the resolution is low, and complex rearrangements are not usually easily detectable. Intrachromosomal rearrangements, which account for about 50% of *FGFR2* fusions in intrahepatic cholangiocarcinoma, can also lead to false-negative results of FISH analysis. In addition, the analysis is mainly restricted to the detection of DNA. FISH analysis, using break-apart probes, has been frequently used to detect *FGFR* fusions in clinical samples [36,44]. Recently, a novel RNA-FISH assay allowed the detection of *FGFR3–TACC3* fusions in bladder cancer [49].

Immunohistochemistry can also detect fusions when rearrangements lead to overexpression of the kinase. Immunohistochemistry is inexpensive and provides information about specific fusions by protein localization, but this approach has a very low sensitivity in identifying rare fusions. So far, no immunohistochemistry method has been proven to have sufficient sensitivity and specificity to detect FGFR fusions [30].

The introduction of NGS technologies, able to identify different types of genomic alterations, including fusions either at DNA or RNA level in a single experiment, allowed the discovery of about the 90% of about 10,000 known gene fusions [4]. Among the NGS analytical strategies, whole-genome sequencing (WGS) has the advantage of identifying a large number of rearrangements and characterize breakpoints, including those in non-coding regions, and it is particularly useful for the discovery of novel fusions. However, this approach is very expensive and time consuming, due to the large quantity of data generated and computational analyses. Whole-exome sequencing (WES) is less suitable than WGS, as it can detect few rearrangements with breakpoints in or near exons [50]. Instead, whole-transcriptome sequencing has been used for the discovery of several gene fusions in different cancer types [32,51]. As compared with WGS, RNA-based testing is more sensitive, efficient, and functionally definitive considering that, although many rearrangements might be present in the genome of tumor samples, only few produce transcripts. A next-generation transcriptome approach was indeed used for the discovery of the first *FGFR* family gene fusion, *FGFR3–TACC3* in glioblastoma multiforme [27]. However, whole-transcriptome analysis is quite expensive, requires personnel with expertise in bioinformatics for data analysis and interpretation, and is not applicable for routine clinical-grade testing.

Targeted sequencing approaches that allow the isolation and sequencing of subsets of genes or regions of the entire genome can detect fusions in a more focused manner. This strategy is a sensitive approach in detecting fusions, as the sequencing coverage is higher than that of WGS. In addition, targeted sequencing is a suitable approach for detecting fusions in clinical diagnostics, as it can investigate DNA and/or RNA and does not require extensive validation of the method, being available several CE-IVD panels (Table 2). DNA-based methods have the advantage that DNA is more stable than RNA, but the detection of novel fusions might be limited, especially when large intronic regions are involved. RNA-based methods are able to distinguish in-frame, transcribed gene fusions versus out-of-frame fusions and avoid difficulties of sequencing large intronic regions. The main weaknesses of RNA-based methods are that the sensitivity depends on the fusion expression level and that RNA is less stable than DNA, especially when FFPE samples are used.

Different approaches are currently available for detecting fusions, i.e., hybrid capture-based methods, amplicon-based approaches, and anchored multiplex PCR (Table 2). Hybridization-capture methods use sequence-specific probes complementary to a specific region of interest that are longer than PCR primers, making it possible to sequence target regions and regions flanking the target. As compared to amplicon-based approaches, hybridization-based strategies are less likely to miss variants, although the detection of fusions might be challenging if introns are long (Figure 1). Amplicon-based enrichment methods use primers specific to known fusion partners and allow the detection of fusions starting from a very low input (≥10 ng) of RNA. This approach can be suitable also for degraded RNA commonly derived from FFPE clinical samples. Other advantages include a simplified workflow, a reduced complexity of data analysis, and a short time for test execution. However, amplicon-based methods limit the discovery of novel fusions, without a prior knowledge of potential partners. A number of FGFR fusions have been detected in clinical samples using these approaches [31,33].

A recent strategy used for target enrichment of RNA libraries for NGS is the anchored multiplex PCR, for example, using the Archer FusionPlex chemistry that uses gene-specific primers anchored to an exon–intron boundary and universal reverse primers that permit the amplification of both known and unknown genomic regions of interest (Table 2 and Figure 1). The main advantage of this technology is the enrichment of a target region with knowledge of only one of its ends [52]. In particular, this method allows the detection of any fusion partner, even if only one of the fusion partners is known, and gives information on the imbalance between 5′ expression and 3′ expression, a phenomenon frequently observed in samples positive for fusions involving a driver gene. Anchored multiplex PCR has been recently used to identify various *FGFR2* fusions in cholangiocarcinoma clinical samples [53].

## 5. Prognostic Significance of *FGFR* Fusions 

The improvement of diagnostic strategies for detection of *FGFR* alterations allowed the identification of a number of *FGFR* fusions that might potentially predict the outcome of cancer patients. The prognostic role of *FGFR* fusions has mainly investigated in biliary tract cancer. In this regard, a study evaluated the presence of *FGFR2* translocations in 152 cholangiocarcinoma and 4 intraductal papillary neoplasms of the bile duct by FISH [54]. Thirteen specimens were positive for *FGFR2* translocations. The median cancer-specific survival interval for patients carrying *FGFR2* translocations was significantly longer (123 months) than that for patients without *FGFR2* translocations (37 months, P = 0.039) [54]. In a study in which 377 patients with biliary tract cancer were enrolled, 95 *FGFR* genetic alterations, including 63 *FGFR2* fusions, were detected. Patients with *FGFR* alterations experienced significantly longer overall survival (OS) than patients without *FGFR* aberrations (37 vs. 20 months; P <0.001) [35]. In a recent study in patients with fluke associated-intrahepatic cholangiocarcinoma, the presence of rare *FGFR2* fusions indicated a trend toward better OS compared with that of fusion-negative tumors, although the difference was not statistically significant [53]. The presence of *FGFR* genomic alterations, including *FGFR2* fusions genes identified by NGS in 55 patients with intrahepatic cholangiocarcinoma, has been associated with an indolent disease course and prolonged survival [55]. However, in this study, *FGFR2* fusions have been reported only in three patients. Interestingly, one patient with an *FGFR2–NOL4* fusion and a co-existing *BAP1* mutation had a rapidly progressive course [55]. In the study of Arai et al. in which seven *FGFR2–AHCYL1*-positive and two *FGFR2–BICC1*-positive intrahepatic cholangiocarcinomas were identified, no significant differences in term of prognosis between fusion-positive and -negative patients were observed [32]. In this study, *KRAS* and *BRAF* mutations were mutually exclusive with *FGFR2* fusions. 

The prognostic significance of *FGFR1–3* fusions was explored in NSCLC. In a study in which 1328 NSCLC patients were enrolled, 17 had *FGFR* fusions (2 *BAG4–FGFR1* and 15 *FGFR3–TACC3*) [30]. No significant differences in relapse-free survival (RFS) or OS were observed between patients with *FGFR*-fusion-positive and *FGFR*-fusion-negative tumors [30].

## 6. FGFR Fusions as Therapeutic Target for Solid Tumors

Even though *FGFR* fusions are rare in human cancers, the field of therapies targeting these molecular alterations has exponentially progressed, thanks to development of a number of novel compounds, including non-selective and selective tyrosine kinase inhibitors (TKIs). 

The first anti-FGFR agents were multi-kinase non-selective inhibitors (e.g., dovitinib, lenvatinib, lucitanib, nintedanib, derazantinib, and ponatinib) that, in addition to inhibit FGFRs, are active against different tyrosine kinases, including VEGFRs, RET, KIT, and PDGFRs, due to the similarity of the intracellular tyrosine kinase domains (Table 3). However, these compounds lack specificity and potency for treatment of FGFR-driven tumors, with an increased risk of adverse events at the doses required for FGFR inhibition. In this regard, the occurrence of several adverse events, including cardiovascular effects, have been reported [56,57,58]. Different clinical trials of non-selective TKIs are ongoing in patients with *FGFR* alterations (Table 3). Few studies reported some activity of these agents in FGFR-driven tumors. In this regard, in a study of dovitinib in 13 patients with Bacillus Calmette–Guerin (BCG)-refractory urothelial carcinoma and *FGFR3* alterations, three patients had *FGFR3* mutations [58]. The response rate (RR) was 8% with only one complete response (CR) in a patient carrying the *FGFR3* S249C mutation. All patients experienced grade 3–4 toxicity [58]. In a clinical study in *FGFR2*-mutant or wild-type endometrial cancer patients treated with dovitinib, the RR in the *FGFR2* mutant group was 5% (11% for all patients); only 1/22 *FGFR2* mutant patients achieved a partial response (PR) [57]. Treatment with derazantinib produced an overall RR (ORR) of 20.7%, a disease control rate (DCR) of 82.8%, and a median progression-free survival (PFS) of 5.7 months in patients with advanced, unresectable intrahepatic cholangiocarcinoma and *FGFR2* fusions who progressed after chemotherapy [59]. In a phase I/II trial of lucitanib, the clinical activity of the drug was evaluated in an expansion cohort of 23 patients with *FGFR*-aberrant tumors. The RR was 30.4% with a PFS of 32.1 weeks [60].

More recently, inhibitors that reversibly or irreversibly bind to the adenosine triphosphate (ATP) pocket of FGFRs and selectively inhibit the activity of the receptors have been developed.

Only two selective FGFR-TKIs have been approved up to now by the Food and Drug Administration (FDA) for the treatment of FGFR-driven cancer. In particular, erdafitinib has been approved for the treatment of patients with locally advanced or metastatic urothelial carcinoma with *FGFR3* or *FGFR2* genetic alterations, including R248C, S249C, G370C, and Y373C mutations and *FGFR3–TACC3* fusions, on the basis of the BLC2001 trial [61]. In this phase 2 study that enrolled 99 patients with advanced urothelial carcinoma carrying *FGFR3* or *FGFR2/3* genomic alterations, an RR of 40%, with 3% of patients obtaining a CR, was observed. In the subgroup of 25 patients with *FGFR* fusions, the RR was 16% [61]. Pemigatinib was granted FDA-accelerated approval in April 2020 for the treatment of cholangiocarcinoma patients with *FGFR2* fusions or rearrangements. The efficacy of the drug was evaluated in the FIGHT-202 study in 107 patients with cholangiocarcinoma and *FGFR2* gene fusions. The ORR was 35.5%, including 3 CRs. No CRs or PRs were observed in patients with other *FGF/FGFR* alterations or no *FGF/FGFR* alterations [62].

A number of different reversible competitive inhibitors directed against multiple FGFRs (e.g., erdafitinib, pemigatinib, infigratinib, rogaratinib, AZD4547, Debio1347) are in clinical development in patients with hematologic and solid tumors who carry *FGFR* alterations (Table 4). FGFR4 selective agents (fisogatinib, H3B-6527, and FGF401) are also under investigation.

The majority of FGFR selective inhibitors are under evaluation in clinical trials in which only patients harboring *FGFR* genomic alterations are enrolled (Table 4). In a phase II trial of AZD4547 in patients with advanced cancers with *FGFR1–3* aberrations, PRs were observed in 4 of 48 (8%) patients, including 2 patients with *FGFR* mutations and 2 with *FGFR3–TACC3* fusions [63]. The estimated median PFS was 3.4 months. The 6-month PFS rate was low for patients with *FGFR* amplifications (0%) and for patients carrying *FGFR* mutations (6%) and higher for patients with *FGFR* fusions (56%). Three of nine patients with *FGFR* fusions had a PFS > 10 months [63]. In a multicenter, open label, phase II study on infigratinib in chemotherapy-refractory advanced or metastatic cholangiocarcinoma with *FGFR* alterations, including 48 *FGFR2* fusions, all responsive cases harbored *FGFR2* fusions [64]. The RR was 14.8 % and the DCR 75.4% (18.8% and 83.3% for patients with *FGFR2* fusions, respectively). Reduced target lesion size in at least one disease evaluation was observed in 36/48 patients with tumors bearing *FGFR2* fusions [64]. 

Only few studies with selective reversible FGFR-TKIs are planned in patients specifically carrying only *FGFR* fusions, presumably due to the low frequency of these alterations (Table 4). 

In this regard, a study exploring the effects of AZD4547 is ongoing in patients with glioma and the *FGFR3–TACC3* fusion (ClinicalTrials.gov Identifier: NCT02824133). Infigratinib is under evaluation in a phase III study as first-line treatment for patients with cholangiocarcinoma and *FGFR2* gene fusions/translocations (ClinicalTrials.gov Identifier: NCT03773302) and in a phase I study in patients with high-grade glioma and *FGFR3–TACC3* translocations (ClinicalTrials.gov Identifier: NCT04424966).

Irreversible inhibitors that covalently bind to a highly conserved P-loop cysteine residue in the ATP pocket of FGFRs have also been developed. Futibatinib is a potent and high selective, irreversible FGFR1–4 inhibitor [65] (Table 4). In a trial exploring futibatinib in previously treated cholangiocarcinoma patients with FGFR alterations, 20/28 patients carrying *FGFR2* fusions experienced tumor shrinkage and 7/28 confirmed PR. The ORR was 25% and the DCR 79% [66]. The irreversible FGFR inhibitor futibatinib is currently under evaluation in a phase III clinical study in patients with advanced cholangiocarcinoma harboring *FGFR2* gene rearrangements (ClinicalTrials.gov Identifier: NCT04093362). 

## 7. Conclusions and Perspectives

Increasing evidence suggests that the FGFR system plays an important role in cancer development and progression. However, the results of clinical trials have clearly demonstrated that only tumors carrying genetic alterations of the *FGFRs* such as mutations or fusions might respond to treatment with FGFR inhibitors, at least when used as single agents. In this respect, targeted therapies for *FGFR*-aberrant tumors may offer an effective therapeutic strategy in some cancer types, such as cholangiocarcinoma, which is often diagnosed in advanced stages when only palliative treatment is available [67]. In the last years, the rapid improvement in the development of drugs targeting *FGFR* alterations, including fusions, combined with the availability of ever more efficient diagnostic tests, allowed the selection of patients who might benefit from FGFR inhibitors. However, some issues should be considered, such as the need of adequate tools for the detection of *FGFR* genetic alterations, the identification of the mechanisms of resistance to FGFR inhibitors and the possibility of performing clinical trials specifically for patients with rare alterations.

*FGFR* fusions are relatively rare genetic alterations. The introduction in clinical diagnostics of NGS panels for comprehensive genomic profiling allowed significant improvement in the detection of *FGFR* alterations [31,33,53]. However, most available targeted sequencing panels for the detection of *FGFR* fusions have some limits as we discussed in this review. In addition, lack of tumor tissue is still a limit to perform genomic profiling of patients with advanced disease. In this respect, analysis of circulating cell-free DNA (cfDNA) might represent an alternative approach in patients with no tissue available for genomic profiling [68]. Different targeted sequencing panels have been developed that allow the analysis of a large number of genes, including *FGFRs*, in cfDNA. Analysis of cfDNA for the detection of *FGFR* fusions might also serve as a non-invasive tool for monitoring patients undergoing FGFR-targeted therapies and for the identification of biomarkers of resistance. However, some technical issues, such as the low recovery of cfDNA, the low fraction of circulating tumor DNA (<0.1% to 50%) in cfDNA, the short half-life, and the high grade of fragmentation, should be resolved for implementing the clinical validity and utility of this approach [69].

Despite the encouraging results of FGFR inhibitors in clinical trials, mechanisms of acquired resistance with the occurrence of secondary mutations have been described, thus limiting the duration of the response. In particular, in three fusion-positive patients with cholangiocarcinoma treated with BGJ398, the emergence of a secondary *FGFR2* kinase domain mutation in one patient and multiple *FGFR2* mutations in the remaining two patients was observed [70]. Interestingly, one *FGFR2* point mutation (p.V564F) was identified in all patients, suggesting a relevant role of this genomic alteration in the resistance to anti-FGFR agents [70]. A recent study in patients with fusion-positive intrahepatic cholangiocarcinoma who progressed on BGJ398 or Debio1347 revealed that treatment with the FGFR irreversible inhibitor futibatinib might overcome the acquired resistance to FGFR reversible inhibitors [71]. Prospective studies in a large population of patients might confirm these findings.

Finally, it is important to consider that *FGFR* fusions occur at low frequency in human tumors [17]. The clinical development of targeted agents directed against rare alterations in a specific tumor type is difficult, due to the small number of patients who can be included in clinical studies. Basket trials, in which a sufficient number of patients with specific genetic alterations can be enrolled, regardless of the tumor type, are required, in order to study the significance of these alterations in a larger population and to offer a personalized treatment to patients carrying these rare genomic aberrations. In this respect, clinical trials are ongoing to explore the agnostic role of *FGFR* fusions as marker of response to drugs targeting the FGFR kinase.

In conclusion, the awareness that *FGFR* alterations, including fusions, play an important role in cancer has greatly enhanced the clinical development of FGFR inhibitors together with the improvement of NGS-based molecular tests. The design of basket trials might significantly improve the approval of FGFR agents for patients carrying *FGFR* fusions.

## Figures and Tables

**Figure 1 ijms-21-06856-f001:**
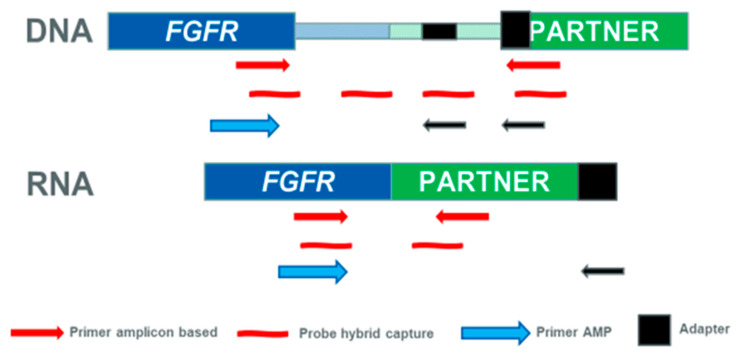
DNA- and RNA-targeted sequencing approaches for detecting FGFR fusions. Hybrid capture methods use sequence specific probes complementary to a specific region of interest that are longer than PCR primers, whereas amplicon-based enrichment methods use primers specific to known fusion partners. The anchored multiplex PCR approach uses gene-specific primers and universal reverse primers that permit the amplification of both known and unknown genomic regions of interest. When DNA is used as starting material, hybrid capture probes can be designed to capture both exons and introns. RNA-based methods detect only functional transcripts, avoiding the difficulties of sequencing large intronic regions.

**Table 1 ijms-21-06856-t001:** Most frequent fibroblast growth factor receptor (FGFR) fusions in solid tumors.

Gene	5′-Gene	3′-Gene	Tumor Type	No. of Cases Reported (Ref.)
***FGFR1***	*FGFR1*	*HOOK3*	GIST	1/186 [25]
	*FGFR1*	*TACC1*	GIST	1/186 [25]
			Glioma	1/795 [26]
			Glioblastoma	1/97 [27]
	*FGFR1*	*ZNF703*	Breast cancer	1/24 [28]
	*FGFR1*	*NTM*	Bladder urothelial carcinoma	1/295 [29]
	*BAG4*	*FGFR1*	Non-small cell lung cancer	2/1328 [30]; 1/26,054 [31]
***FGFR2***	*FGFR2*	*AHCYL*	Cholangiocarcinoma	7/102 [32]
	*FGFR2*	*BICC1*	Cholangiocarcinoma	2/102 [32]; 6/195 [34]; 8/377 [35]; 40/107 [37];
			Colorectal cancer	1/149 [32];
			Hepatocarcinoma	1/96 [32]
	*FGFR2*	*PPHLN1*	Cholangiocarcinoma	16/107 [37]
	*FGFR2*	*TACC3*	Cholangiocarcinoma	1/6 [36]
	*FGFR2*	*CCDC6*	Cholangiocarcinoma	3/377 [35]
	*FGFR2*	*KIAA1598*	Non-small cell lung cancer	2/26054 [31]
***FGFR3***	*FGFR3*	*TACC3*	Glioblastoma	2/97 [27]
			Glioma	20/795 [26]
			Non-small cell lung cancer	15/1328 [30]; 37/26,054 [31]
			Bladder cancer	3/2375 [38]
			Head and neck squamous cancer	2/2375 [38]
			Lung squamous cell carcinoma	4/2375 [38]
	*FGFR3*	*BAIAP2L1*	Bladder cancer	1/2375 [38]; 2/46 [44]
			Lung cancer	2/83 [44]

GIST, gastrointestinal stromal tumor.

**Table 2 ijms-21-06856-t002:** Commercially available targeted sequencing kits for fusion detection.

Technology	Kit	Sample	Nucleic Acid	Input	No. of Genes	No. of Fusion Genes
**Hybrid Capture-based**	FoundationOneCDx (Foundation Medicine)	FFPE	DNA	Moderate (≥50 ng FFPE RNA)	324	36, including *FGFR1–3*
	TruSight Tumor 170 (Illumina)	FFPE	DNA/RNA	Moderate (≥40 ng FFPE DNA/RNA)	170	55, including *FGFR1–4*
	TruSight Oncology 500 (Illumina)	FFPE	DNA/RNA	Moderate (≥40 ng FFPE DNA/RNA)	523	55, including *FGFR1–4*
**Amplicon-based**	Oncomine comprehensive assay (Thermofisher)	FFPE	DNA/RNA	Low (≥10 ng FFPE DNA/RNA)	161	51, including *FGFR1–3*
	Oncomine Focus Assay (Thermofisher)	FFPE	DNA/RNA	Low (≥10 ng FFPE DNA/RNA)	52	23, including *FGFR1–3*
**Anchored multiplex PCR-based**	FusionPlex Oncology Research(ArcherDX)	Fresh, frozen, and FFPE	RNA	Moderate (≥50 ng ^§^ FFPE RNA)	75	75^¥^, including *FGFR1–3*
	FusionPlex Solid Tumor (ArcherDX)	Fresh, frozen, and FFPE	RNA	Moderate (≥50 ng ^§^ FFPE RNA)	53	53 ^¥^, including *FGFR1–3*
	FusionPlex Comprehensive Thyroid and Lung (CTL) (ArcherDX)	Fresh, frozen, and FFPE	RNA	Moderate (≥50 ng ^§^ FFPE RNA)	36	16 ^¥^, including *FGFR1–3*
	FusionPlex Lung (ArcherDX)	Fresh, frozen, and FFPE	RNA	Moderate (≥50 ng ^§^ FFPE RNA)	14	13 ^¥^, including *FGFR1–3*

Abbreviations: FFPE formalin-fixed, paraffin-embedded; ^§^ recommended input in the absence of PreSeq screening; ^¥^ fusion, splicing, or exon-skipping.

**Table 3 ijms-21-06856-t003:** Clinical trials of non-selective FGFR inhibitors in patients with solid tumors and *FGFR* genetic alterations.

Compound	Target	Eligibility on the Basis of *FGFR* Alterations	Tumor Type	Phase	ClinicalTrial Identifier
**Dovitinib**	FGFR1–2–3; VEGFR1–2–3; PDGFRβ	*FGFR3* mutation/over-expression	BCG refractory urothelial carcinoma	II	NCT01732107
		*FGFR* mutation/ amplification/ translocation	Solid and hematologic tumors	II	NCT01831726
		*FGFR2* amplification	Metastatic or unresectable gastric cancer	II	NCT01719549
		*FGFR2* mutation or FGFR2 wild type	Advanced and/or metastatic endometrial cancer	II	NCT01379534
**Lucitanib**	FGFR1–2–3; VEGFR 1–2–3; PDGFRα-β; CSF1R	*FGFR 1–3* gene fusion/activating mutation	Advanced/metastatic lung cancer	II	NCT02109016
		*FGFR* aberrations	Advanced cancers	II	NCT02747797
		*FGFR1* amplification or *FGFR1* wild type	Estrogen receptor-positive metastatic breast cancer	II	NCT02053636
**Nintedanib**	FGFR1–2–3; VEGFR 1–2–3; PDGFRα-β	*FGFR 1–3* alterations	Advanced non-small cell lung cancer	II	NCT02299141
		*FGFR3* mutation/ overexpression or *FGFR3* wild type	Advanced urothelial carcinoma	II	NCT02278978
**Ponatinib**	FGFR1, VEGFR2; BCR–ABL, SRC; KIT; PDGFRα	*FGFR* mutation/ fusion/amplification	Advanced cancers	II	NCT02272998
		*FGFR2* fusion	Advanced biliary cancer	II	NCT02265341
**Derazantinib**	FGFR1–3, CSF1R, RET; KIT; PDGFRβ	*FGFR* aberrations	Advanced urothelial cancer	I/II	NCT04045613
		*FGFR* genetic alterations	Advanced solid tumors	I/II	NCT01752920
		*FGFR2* fusion/ mutation/amplification	Inoperable or advanced intrahepatic cholangiocarcinoma	II	NCT03230318

BCG, Bacillus Calmette–Guerin.

**Table 4 ijms-21-06856-t004:** Clinical trials with selective FGFR inhibitors.

Drug	Target	Tumor Type	Phase	Status	Clinical Trial Identifier
**Reversible FGFR inhibitors**					
**Erdafitinib (JNJ-42756493)**	FGFR1–4	*FGFR*aberrant advanced refractory solid tumors, lymphomas, or multiple myeloma	II	recruiting	NCT02465060
		*FGFR*-aberrant urothelial cancer	II	active	NCT02365597
		*FGFR*-aberrant advanced squamous non-small-cell lung carcinoma	II	recruiting	NCT03827850
		*FGFR*-aberrant urothelial cancer	III	recruiting	NCT03390504
		*ER+/HER2-/FGFR*-amplified metastatic breast cancer	I	recruiting	NCT03238196
		*FGFR*-aberrant advanced non-small-cell lung cancer, urothelial cancer, esophageal cancer, or cholangiocarcinoma	II	recruiting	NCT02699606
		Advanced solid tumor with *FGFR* mutation or gene fusion	II	recruiting	NCT04083976
**Pemigatinib (INCB054828)**	FGFR1–3	*FGFR*-aberrant advanced solid malignancies	I/II	recruiting	NCT02393248
		*FGFR3* mutant or rearranged metastatic or unresectable urothelial carcinoma	II	recruiting	NCT04003610
		*FGFR2* rearranged unresectable or metastatic cholangiocarcinoma	III	recruiting	NCT03656536
		*FGFR*-aberrant unresectable advanced, relapsed, or metastatic solid tumors	I	not yet recruiting	NCT04258527
		*FGFR*-aberrant metastatic or unresectable colorectal cancer	II	not yet recruiting	NCT04096417
		*FGFR*-aberrant metastatic or surgically unresectable urothelial carcinoma	II	recruiting	NCT02872714
		Locally advanced/metastatic or surgically unresectable solid tumor malignancies with activating *FGFR* mutations or translocations	II	recruiting	NCT03822117
		High-risk patients with urothelial carcinoma with activating *FGFR* mutations or translocations	II	not yet recruiting	NCT04294277
**Infigratinib (BGJ398)**	FGFR1–3	Advanced or metastatic cholangiocarcinoma with *FGFR2* gene fusions or translocations or other *FGFR* genetic alterations	II	recruiting	NCT02150967
		Invasive urothelial carcinoma and *FGFR3* genetic alterations	III	recruiting	NCT04197986
		Advanced or metastatic solid tumors with *FGFR1–3* gene fusions or other *FGFR* genetic alterations	II	Recruiting	NCT04233567
		Unresectable locally advanced or metastatic cholangiocarcinoma with *FGFR2* gene fusions/translocations	III	recruiting	NCT03773302
		Recurrent high-grade glioma with *FGFR1* K656E or *FGFR3* K650E mutation or FGFR3–TACC3 translocation	I	recruiting	NCT04424966
**Rogaratinib (BAY1163877)**	FGFR1–4	Recurrent or metastatic squamous cell carcinoma of the head and neck with *FGFR1/2/3* mRNA overexpression	II	recruiting	NCT03088059
		Metastatic gastric cancer with *FGFR* mutation/fusion	II	not yet recruiting	NCT04077255
		*FGFR*-positive locally advanced or metastatic urothelial carcinoma	II/III	active, not recruiting	NCT03410693
		*FGFR1–3*-positive advanced squamous-cell non-small cell lung cancer	II	recruiting	NCT03762122
**AZD4547**	FGFR1–3	*FGFR*-aberrant advanced refractory solid tumors, lymphomas, or multiple myeloma	II	recruiting	NCT02465060
		Muscle-invasive bladder cancer with *FGFR* mutations/fusions	I	active not recruiting	NCT02546661
		*ER+* breast cancer patients with *FGFR1* polysomy (FISH4/5) or gene amplification	I/II	completed	NCT01202591
		*FGFR*-aberrant squamous cell lung cancer	II/III	active not recruiting	NCT02965378
		*FGFR1*-amplified squamous non-small-cell lung cancer	I/II	completed	NCT01824901
		Recurrent malignant glioma expressing *FGFR–TACC* Gene Fusion	I/II	completed	NCT02824133
		Advanced refractory cancers/lymphomas/multiple myeloma	II	active not recruiting	NCT04439240
**DEBIO 1347 (CH5183284)**	FGFR1–3	Solid tumors harboring a fusion of *FGFR1, FGFR2,* or *FGFR3*	II	active not recruiting	NCT03834220
		*FGFR*-amplified metastatic breast cancer	I/II	recruiting	NCT03344536
		*FGFR*-aberrant advanced solid tumors	I	active not recruiting	NCT01948297
**Irreversible FGFR inhibitors**					
**Futibatinib (TAS-120)**	FGFR1–4	*FGFR*-aberrant advanced or metastatic solid tumor, advanced or metastatic gastric or gastroesophageal cancer, myeloid or lymphoid neoplasms	II	not yet recruiting	NCT04189445
		Advanced cholangiocarcinoma harboring *FGFR2* gene rearrangements	III	not yet recruiting	NCT04093362
		*FGF/FGFR* aberrant advanced solid tumors	I/II	active, not recruiting	NCT02052778

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
