# Peer review of "FGFR Fusions in Cancer: From Diagnostic Approaches to Therapeutic Intervention"

_ijms, 2020, doi:10.3390/ijms21186856_

Round 1

Reviewer 1 Report

- explain OS on line 282

- explain PFS on line 317

- move "disease control rate" from line 350 to line 317

Reviewer 2 Report

In this review, the authors want to describe FGFR aberrations in human cancer. By focusing on FGFR fusion proteins and the potential role of these aberrations as prognostic markers.
The topics are very well developed and treated in an orderly and organic way.
However, I would advise the authors to give more weight to a much discussed topic in recent years, namely the impact of aberrant splicing of the various FGFR isoforms which has been seen to have important effects on tumorigenesis. For example, there are papers indicating aberrant splicing induced by oncovirus proteins that drive towards EMT. Or there are oncogenic effects of aberrant splicing even in normal cellular contexts, Etc. All to underline that fusion proteins can also have an impact on the splicing of FGFR isoforms and therefore on tumor evolution and development.
